# Impacts of Zika emergence in Latin America on endemic dengue transmission

Rebecca K. Borchering[1,2]*, Angkana T. Huang [1], Luis Mier-y-Teran-Romero[3], Diana P. Rojas[4], Isabel Rodriguez-Barraquer[5], Leah C. Katzelnick [1], Silvio D. Martinez[1], Gregory D. King [1], Stephanie C. Cinkovich[1], Justin Lessler[3] & Derek A.T. Cummings [1]*

In 2015 and 2016, Zika virus (ZIKV) swept through dengue virus (DENV) endemic areas of Latin America. These viruses are of the same family, share a vector and may interact competitively or synergistically through human immune responses. We examine dengue incidence from Brazil and Colombia before, during, and after the Zika epidemic. We find evidence that dengue incidence was atypically low in 2017 in both countries. We investigate whether subnational Zika incidence is associated with changes in dengue incidence and find mixed results. Using simulations with multiple assumptions of interactions between DENV and ZIKV, we find cross-protection suppresses incidence of dengue following Zika outbreaks and low periods of dengue incidence are followed by resurgence. Our simulations suggest correlations in DENV and ZIKV reproduction numbers could complicate associations between ZIKV incidence and post-ZIKV DENV incidence and that periods of low dengue incidence are followed by large increases in dengue incidence.

[1] Department of Biology and Emerging Pathogens Institute, University of Florida, Gainesville, FL 32610, USA. [2] Odum School of Ecology, University of Georgia, Athens, GA 30602, USA. [3] Department of Epidemiology, Johns Hopkins Bloomberg School of Public Health, Johns Hopkins University, Baltimore, MD 21205, USA. [4] Department of Biostatistics, College of Public Health and Health Professions, University of Florida, Gainesville, FL 32610, USA. [5] Department of Medicine, University of California, San Francisco, San Francisco, CA 94143, USA. *email: rebecca.borchering@uga.edu; datc@ufl.edu

In 2015 and 2016, Zika virus (ZIKV) swept through many Latin American countries[1] where dengue virus (DENV) is endemic. Following this epidemic, many locations appeared to experience abnormally low dengue incidence. DENV and ZIKV share a vector[2,3] and are both flaviviruses. Evidence suggests these viruses may interact competitively or synergistically through human immune responses: via antibodies in the case of non-overlapping infections[4–10] or innate defenses during co-infections[11,12]. Concurrent infections in the vector could also potentially alter viral fitness though the low prevalence of infection in mosquitos at any time and thus the low rate at which concurrent infections occur is likely to minimize the impact of this interaction[12–14]. It is also possible that these viruses may have no biological interaction whatsoever. Changes in surveillance and control in response to Zika[1,15] could affect reported dengue cases. Conditions (climate[16], vector abundance) that favor Zika may be similar to those favoring dengue.

Epidemics of emerging pathogens have the potential to disrupt the ecology of other circulating pathogens. ZIKV was identified in the Americas in Brazil in late 2015[17], though phylogeographic analyses suggest ZIKV may have arrived as early as mid-2013[18], entering northeast Brazil in early 2014[19]. The outbreak in the Americas was extensive, affecting 48 countries and territories up through December 2017[20]. Serological evidence has found rates of infection ranging from 56% (Nicaragua[21]) to 63% (Salvador, Brazil[22]). These are comparable to seroprevalence in previous ZIKV outbreaks; 73% in Yap Island, Micronesia[23], 49% in French Polynesia[24]. Associations between ZIKV infection and severe disease outcomes such as microcephaly and Guillain-Barré syndrome were recognized by the WHO in March 2016[1], prompting intensified surveillance and control efforts.

Since 2016, Zika incidence in Brazil has dropped precipitously, from over 200,000 probable cases in 2016 to 18,548 in 2017[25]. In Colombia incidence dropped from approximately 90,000 in 2016 to 1641 in 2017[25]. These reductions are likely the result of widespread immunity throughout affected populations, leaving few individuals susceptible to infection. Multiple human and animal studies show that ZIKV induces potently neutralizing antibody responses[26,27], suggesting enduring ZIKV-specific immunity[28].

The primary ZIKV vectors, Aedes aegypti and Aedes albopictus[2,3,29], also transmit chikungunya virus (CHIKV) and DENV. In contrast to CHIKV (an alphavirus), DENV and ZIKV are genetically similar flaviviruses[8]. DENV exists as four antigenically distinct serotypes (DENV1-4). When an individual is infected by DENV, there is a period of cross-protection from infection by other serotypes[30]. After the period of cross-protection, subsequent infections with different serotypes can result in more severe disease due to antibody dependent enhancement (ADE)[31,32]. It has been speculated that, due to the similarity between DENV and ZIKV, immunity to one of these viruses may alter the chance of infection or probability of severe disease following exposure to the other virus. Current evidence supports both the potential for enhancement[4–7] and for cross-protection[5,7–10].

Here, we examine data from Brazil (1999–2017)[25,33,34] and Colombia (2007–2017)[25,35,36] to determine whether dengue incidence has been atypical since the emergence of ZIKV. We demonstrate that dengue incidence was significantly lower than expected in both countries in 2017. Despite these unprecedented low periods, we do not find a negative association between cumulative Zika incidence and biweekly dengue incidence in either country. To gain insight into how immunological interactions would impact the relationship between ZIKV and DENV incidence in this period, we use a stochastic compartmental model of the four DENV serotypes and ZIKV under multiple assumptions. In almost all simulations incorporating strong ZIKV cross-protection against subsequent DENV infection, ZIKV epidemics are followed by a trough in dengue incidence, followed by a larger than average peak in DENV incidence. Correlated hazards of DENV and ZIKV transmission may complicate the relationship between ZIKV incidence and resulting dengue incidence. Our simulations consistently show that periods of low dengue incidence are followed by large increases in dengue incidence.

## Results

**Departures from expected dengue incidence.** At the population level, the large ZIKV epidemic could plausibly have led to either increases or decreases in dengue cases. To determine if either was the case, we compared dengue surveillance data from before, during, and after the arrival of ZIKV. We constructed time series of probable case counts for each state in Brazil (1999–2017)[25,33,34] and department in Colombia (2007–2017)[25,35,36] (see "Methods", Fig. 1b, d, Supplementary Fig. 1). In 2017, Brazil had the lowest annual incidence rate (IR) of dengue since 2005 (Fig. 1a) and Colombia had the lowest annual dengue incidence since 2007 (the first year of available data) (Fig. 1c).

We quantified the probability of departures arising by chance from expected dengue incidence using time series models with seasonal variation in autocorrelative effects (see Methods and Supplementary Figs. 2–4 for further details on model implementation and predictive ability). Biweeks with atypically large incidence occurred significantly more often than expected in Brazil in 2015 (Fig. 2a, b). In 2017, Brazil experienced an increase in both atypically high and atypically low biweeks compared to expectation. In Colombia, a significant increase in atypically low biweeks was observed in 2016 and 2017 (Fig. 2c, d) (see "Methods" and Supplementary Figs. 5 and 6 for further details).

**Characterizing departures with hierarchical models.** To attempt to explain the significant departures that we saw in each location, we built a set of hierarchical time series models that incorporated a number of subnational covariates. These models forecasted dengue incidence using combinations of seasonal dengue terms, year effects, and recent incidence of Zika. For a baseline, we also fit models incorporating chikungunya incidence. CHIKV shares climate and vector determinants as DENV and ZIKV but is of a different viral family and thus is not expected to interact immunologically with flaviviruses.

In models incorporating year effects, we found that biweekly expected dengue incidence in 2015 was higher than that of corresponding biweeks in other years in both Brazil (mean: 2.21-fold increase, 95% CrI: 1.45–3.37) and Colombia (mean: 1.11-fold increase, 95% CrI: 0.96–1.29). In contrast, we found that biweekly dengue incidence in 2017 was lower in both Brazil (relative incidence mean: 0.63, 95% CrI: 0.46–0.86) and Colombia (relative incidence mean: 0.23, 95% CrI: 0.18–0.29). At the subnational-level there are differences in the significance and direction of these effects (Fig. 3b, d), with the exception of 2017 in Colombia, where there is a significant reduction in biweekly dengue incidence in all departments (Fig. 3d).

We found a positive association between the previous biweek's Zika incidence and dengue transmission potential in Brazil (mean coefficient: 0.16, 95% CrI: 0.05–0.26, Fig. 3a). Putting this coefficient in context, 1000 Zika cases observed in the previous biweek translates to a multiplicative increase of about three times as many expected dengue cases in the following biweek (mean: 2.92, 95% CrI: 1.40–6.15). We also found a positive association between cumulative Zika incidence on dengue transmission potential in Brazil (mean coefficient: 0.04, 95% CrI: 0.003, 0.08).

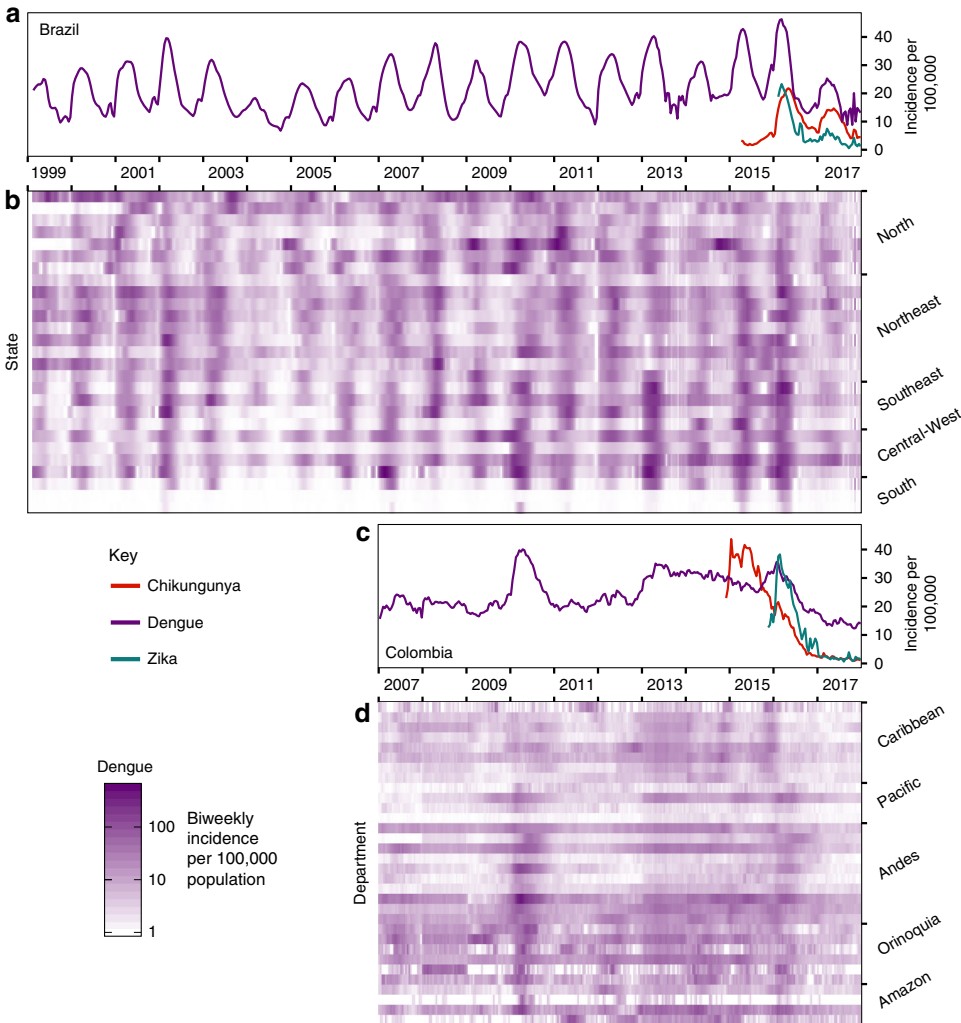

**Fig. 1 Estimated dengue incidence in Brazil and Colombia (per 100,000).** Note that Zika incidence was not systematically reported prior to 2016 in Brazil (**a**) or late 2015 in Colombia (**c**) and that Chikungunya was not systematically reported prior to 2015 and late 2014 respectively in Brazil (**a**) and Colombia (**c**). In Brazil, updated data from the following year's bulletin for 2014 to Epiweek 42, 2017 is used. States in Brazil (**b**) and departments in Colombia (**d**) are arranged by region and then by latitude from North to South.

No significant associations were found between Zika incidence and dengue transmission potential in Colombia.

In Brazil, cumulative (mean coefficient: 0.04, 95% CrI: 0.01–0.08, Fig. 3a) and biweekly (mean coefficient: 0.11, 95% CrI: 0.04–0.17, Fig. 3a) chikungunya incidence were positively associated with expected dengue transmission in the following biweek (Fig. 3a, c, Supplementary Fig. 7). No significant association were found between chikungunya incidence and dengue transmission potential in Colombia.

In Brazil, states with positive effects of Zika on dengue transmission potential often observed positive effects of chikungunya on dengue (Supplementary Fig. 7). Totals of suspected Zika and suspected chikungunya cases at state- and department-level were positively correlated (Supplementary Fig. 8), consistent with potentially shared environmental suitability conditions for transmission of these viruses.

We tested whether the direction of country-level effects was an artifact of our model implementation, by replacing 2015 to 2017 with a random three consecutive years of data preceding 2015 and then repeating the model fitting procedure. We did not find agreement in the directionality of the country-level effects across the resulting models (see Supplementary Fig. 9), supporting our main results.

**Simulations incorporating immune-mediated interactions**. We tested whether immune-mediated interactions between DENV and ZIKV could produce the dengue dynamics observed in Brazil and Colombia by simulating the arrival of Zika in a dengue endemic population (Fig. 4 and Supplementary Figs. 10–12). We used a stochastic compartmental model that incorporated combinations of cross-protection or enhancement between the two viruses (see "Methods" for further details). We performed simulations in which ZIKV was introduced to a population in which DENV was in a stable state as well as simulations that incorporated the sequential introduction of dengue over the decades preceding the ZIKV introduction reflecting the observed detection of DENV serotypes[37] (see Supplementary Fig. 12 for sample simulations from both settings). In simulations where ZIKV infection temporarily reduces an individual's risk of DENV infection by 80%, Zika epidemics are followed by a trough in dengue incidence ranging from 2.2 years to 3.5 years depending on the enhancement scenario (Fig. 4i–l and Supplementary Table 1). Even in the absence of enhancement between DENV and ZIKV, multiple simulations showed increases in dengue after troughs ranging from a 1.3-fold increase to a 2.7-fold increase (Fig. 4). Based on our simulations, which assumed that cross-protection lasts one year on average, the time until dengue

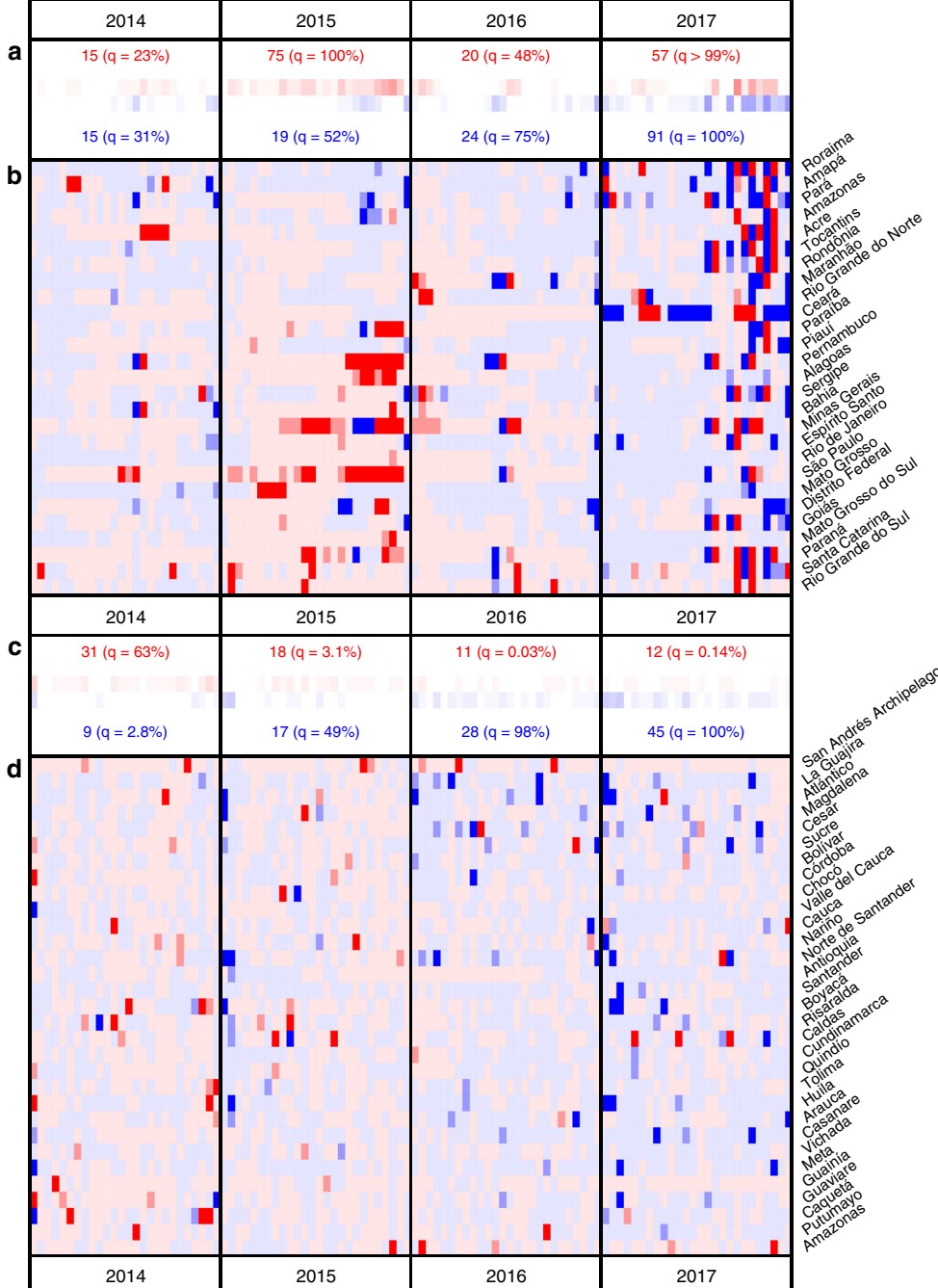

**Fig. 2 Comparison between predicted and observed dengue incidence.** Results for 2014–2017 are shown for Brazil (**a**, **b**) and Colombia (**c**, **d**) (see Supplementary Fig. 5 and 6 for full time series). **b**, **d** Red or blue indicate that the observed incidence fell above or below the median of 500 draws from the posterior of predicted values for that biweek. Medium or dark shading indicates that the observed incidence fell outside of the 90 or 95% prediction interval (PI) for that biweek. **a**, **c** The number of biweeks with observations falling below (blue) or above (red) the 90% PI are displayed with a quantile of the observed number of significant biweeks out of a distribution generated by 10,000 bootstrapped replicates. In these replicates, year labels were randomly re-assigned for each location before counting the biweeks in each year that were above or below the 90% PI (see "Methods" for further details).

resurgence (trough duration) would likely be longer than the assumed one-year duration of cross-protection. In all scenarios, suppression of DENV transmission resulted in subsequent increases in DENV prevalence, suggesting that the low period of incidence observed in 2017 may be followed by large increases in DENV.

We also tested the impact of correlation in ZIKV and DENV transmission intensity on observed associations between cumulative ZIKV and DENV incidence (see Supplementary Fig. 10).

We consistently observed a reduced impact of ZIKV on dengue in simulations where DENV transmissibility was assumed to be higher. When the transmissibility of ZIKV and DENV were assumed to be equal, reductions in DENV due to cross protection were not larger in simulations with higher ZIKV attack rates as the impact of ZIKV was offset by increased DENV transmissibility. These results are consistent with an unclear or variable relationship between DENV reductions and cumulative ZIKV incidence as we have observed in the data.

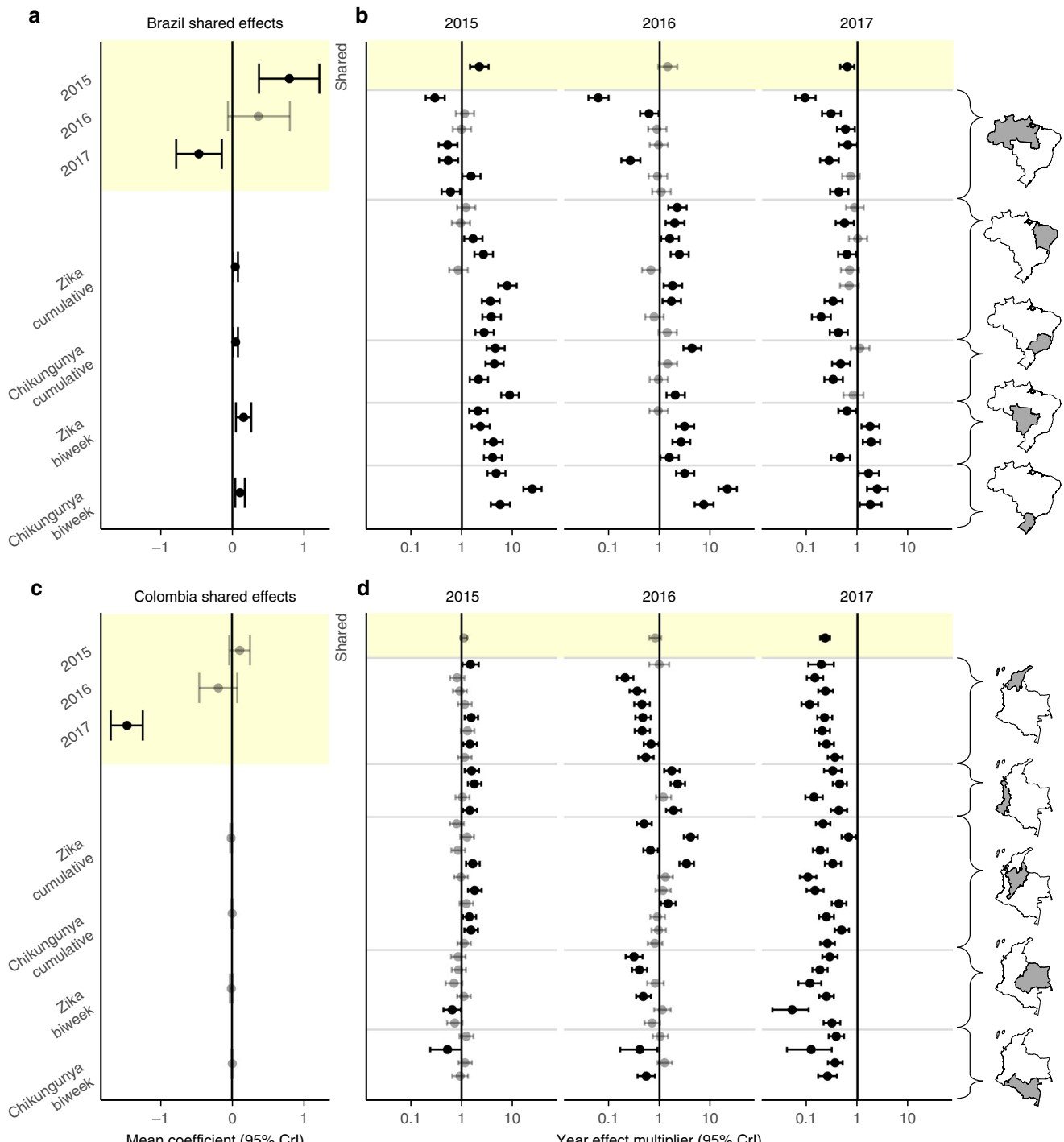

**Fig. 3 Spatial hierarchical biweekly dengue incidence models.** Shared coefficients from the year model are highlighted in yellow. Shared effect coefficients for Brazil (**a**) and Colombia (**c**). Zika and chikungunya coefficients are estimated from autoregressive dengue models. Positive (negative) coefficients indicate increases (decreases) in expected dengue incidence for the year model and indirectly as effects on transmission for the Zika and chikungunya models. Shared year multipliers for expected dengue incidence are shown for Brazil (**b**) and Colombia (**d**). The top row of panels b and d are translated from coefficients in (**a**) and (**c**). Other rows display subnational effects (combined shared and location-specific effects). Mean and 95% credible intervals (CrI) are shown. Intervals that overlap zero are displayed in gray.

## Discussion

In 2017, low dengue incidence rates (Fig. 1), atypically high numbers of biweeks with lower than expected dengue incidence (Fig. 2), and negative country-level effects (Fig. 3a and c) indicate a reduction in dengue incidence in Brazil and Colombia. Utilizing available Zika case data, we were unable to establish a direct link between this reduction and the Zika epidemic. It is important to

note that there are limitations to using passive surveillance data (as we have done here), particularly when novel pathogens are involved. In the case of ZIKV invasion in the Americas, many cases that occurred early in the epidemic were not reported, since it took time to first identify the presence of ZIKV and then to establish reporting protocols. The unavailability of Zika case count data during the height of the ZIKV epidemic in northeast

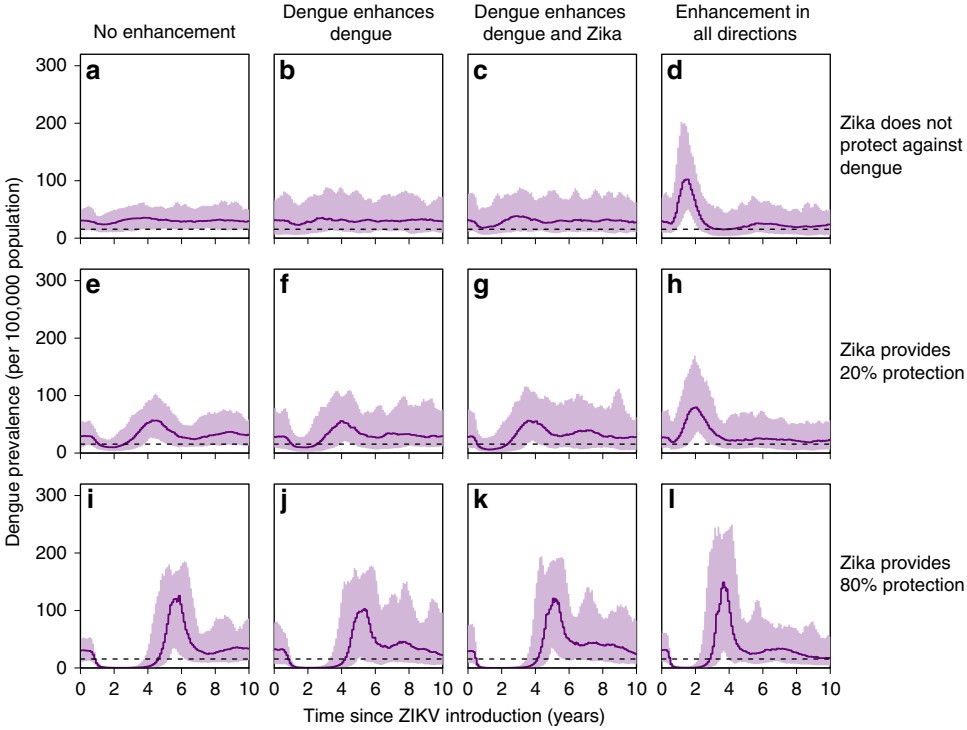

**Fig. 4 Simulation results incorporating immune-mediated interactions between DENV and ZIKV.** Mean and 95% inter-quantile range from stochastic simulations spanning 10 years post ZIKV-introduction. 100 simulations per scenario (see "Methods" for further details). ZIKV introduced after a 100 year burn-in period for four DENV serotypes (see Supplementary Fig. 11 for results when ZIKV is introduced 40 years after DENV). DENV and ZIKV reproduction numbers are assumed to be 4 and 2, respectively. Other reproduction number combinations are considered in Supplementary Fig. 10). The dashed line indicates one-half of the average incidence in (**a**) which we use to define the start and end of DENV prevalence troughs (see Methods and Supplementary Table 1). **a**–**d** Individuals with previous dengue exposure experience 20% of the DENV force of infection (FOI) that a fully susceptible person would. **e**–**h** Individuals with previous ZIKV exposure experience 80% of the FOI that a fully susceptible person would. **i**–**l** Individuals with ZIKV exposure experience 20% of the DENV FOI (same amount of cross-protection between dengue and Zika than between dengue serotypes).

Brazil, one of the most severely affected regions[38], may have contributed to our inability to identify a relationship between cumulative Zika cases and observed dengue incidence. In both Brazil and Colombia, misclassified cases (potentially resulting from shared symptoms between case definitions of dengue, Zika, and chikungunya) also may have restricted our ability to detect such a relationship. Other factors such as climatological effects[16] or additional mosquito control efforts[1,15] and avoidance behavior may have played a role in reducing dengue incidence.

Our results demonstrate a significant decline in dengue cases in 2017 in Brazil and Colombia. Studies involving laboratory confirmed cases and enhanced serosurveillance (for example Ribeiro et al.[38]) will play an important role in pinpointing the mechanism underlying this reduction. Our simulation results show that troughs in dengue incidence are followed by atypically high dengue levels. Atypically low dengue case counts observed in 2017 in Brazil and Colombia suggest that population-level susceptibility to symptomatic dengue has been building. Such high levels of susceptibility could fuel large dengue epidemics in upcoming seasons. As of August, 2019, both Brazil and Colombia have reported more cases at this point in the year than in all of 2017 and 2018, though their dengue seasons are not complete[39]. These early indications are consistent with our expectations of the impact of higher levels of susceptibility due to lower incidence in recent years.

## Methods

**Data sources.** We collated data from Brazil (27 states) and Colombia (32 departments) into a public data repository[25]. For Brazil, monthly dengue data is available for some states starting in 1986 from the National Health Foundation-

FUNASA and for all 27 states starting in 1999. We used monthly data from 1999 to 2012. Monthly data for 2001–2012 were taken from the centralized System for Reporting of Notifiable Conditions (SINAN)[33]. For 2013–2017, we entered weekly data found in the Epidemiological Bulletins[34] published by the Secretariat of Health Surveillance in the Ministry of Health, Brazil where the number of probable cases, severe dengue, dengue with alarm symptoms, and dengue deaths were reported. Severe dengue and dengue with alarm symptoms cases are clinically diagnosed based on symptoms. Case definitions have changed over time, but we use probable case counts (including both confirmed and unconfirmed cases) in all of our analyses. Zika and chikungunya case counts were first reported in the Epidemiological bulletins for Epidemiological week (Epiweek) 13, 2016 and Epiweek 9, 2016 respectively.

Data entry accuracy was checked by re-entering 10% of the weekly bulletins. Within these bulletins, we found that less than 0.01% of the numeric fields entered were inconsistent between the primary and secondary entries.

The Epidemiological Bulletins published in Brazil for 2014–2017 also report the corresponding dengue data for the previous year. We use the available previous year data reported in the 2014–2018 bulletins to obtain probable dengue case counts for 2013–2017 Epiweek 42 since these data are most comparable to the preceding historical data. Analogous updated probable Zika and chikungunya case counts were first reported in 2017 and 2016 respectively. We use these updated counts in our analyses. We also use data on the cumulative number of microcephaly cases and other central nervous system disorders in newborns[40] reported in Brazil since Epiweek 45, 2015 through the end of 2016. Cumulative number of cases later disregarded from the system was also reported. We therefore deduct the disregarded counts from the case counts.

For Colombia, we use weekly department-level probable dengue case count data from the Colombian Instituto Nacional de Salud (INS) website[35]. This dataset includes weekly dengue case counts for 2007–2017, with severe dengue cases reported separately for 2014–2017. We excluded cases with an unknown department and those that were considered imported from other countries. We combined counts for districts with the counts for the department in which they are located, whenever they were reported separately. We extracted additional data for probable Zika and chikungunya cases from the weekly Epidemiological Bulletins[36] for 2015–2017. According to the Epidemiological Bulletins published by Colombia's Directorate of Surveillance and Risk Analysis in Public Health, Zika has been in the country since 2015 Epiweek 40 with the number of cases first reported

in the 2016 Epiweek 1 bulletin. In the updated Colombia dataset, the first documented Zika cases were in 2015 Epiweek 25. Case counts in the bulletins of both countries were given as cumulative counts.

We also gathered subnational level population size data for both Brazil and Colombia[25] from the Instituto Brasileiro de Geografia e Estatística[41] and Departamento Administrativo Nacional de Estadística Colombia[42] respectively.

**Incidence time-series construction.** We construct biweekly time series for probable dengue, Zika, and chikungunya case counts for each state in Brazil and department in Colombia. The Epidemiological Bulletins provide estimates for the cumulative number of cases since Epiweek 1 reported up until the corresponding Epiweek. These cumulative counts are not fixed and are updated as new information becomes available, such as updated knowledge of clinical symptoms or related test results. When the diagnosis of a case is changed to another disease or when samples are negative for the originally designated pathogen, these cases are removed from the cumulative counts. Thus, the published cumulative counts sometimes decrease in consecutive bulletins.

To avoid biweekly incidence values, we constructed strictly nondecreasing upper, middle, and lower approximations of the cumulative time series. Working backward from the most recent time point, we accounted for decreases in the cumulative count by either subtracting the difference from all previous time points (lower approximation) or by setting the higher incidence equal to the lower incidence in the following biweek (upper approximation). This process was repeated for all time points, resulting in a strictly nondecreasing time series. The middle approximation is constructed by averaging the values of the upper and lower approximation at each time point.

To determine which approximation to use in our analyses, we calculated the sum of squared errors for each of the approximations using the updated previous year data (2014–2016) reported in the 2015–2017 Epidemiological Bulletins from Brazil. We found that the middle approximation best represented our target time series; this approximation was thus used for the construction of all case count time series for both Brazil and Colombia. For Brazil, we use the final (updated previous year) data to construct our time series, when it is available. When performing our analysis, we used updated data for Brazil up through Epiweek 42, 2017 (since updated data for the rest of 2017 was unavailable).

For each location and disease, we fit a spline to the adjusted cumulative incidence curve by using the smooth.spline function from the stats package in R. A knot was assigned to each data point. We used a binning procedure to translate the continuous spline into a biweekly sequence of predicted case counts. Cases were assigned to biweeks with probabilities based on the difference in predicted case counts at endpoints of consecutive biweeks. When consecutive predicted spline values decreased, the probability of assigning cases to the first biweek was set to zero. We repeated the reassignment process 1000 times to create 1000 case count time-series and then calculated the mean incidence for each biweek (averaged over the 1000 simulations), rounded down to the nearest integer to obtain the final biweekly case count. In what follows, we refer to this value as the observed incidence.

**Seasonality evaluation.** For each dengue season, we identified the biweeks with the three greatest numbers of cases in each subnational location (states in Brazil and departments in Colombia). In Brazil, dengue seasons were defined to range from biweek 18 to biweek 17 of the next year, to avoid splitting the season into multiple years. In Colombia, there was not a clear start and end of a dengue season, so we did not define a particular season that spans across multiple calendar years. See Supplementary Fig. 1 which describes the seasonality for states and regions.

**Time series models.** For each year of available data, we fit a seasonal one-step autoregressive model[43] with negative binomial errors for each state in Brazil and each department in Colombia using incidence data from that location in all other years. For each location, the number of dengue cases in the following biweek is modeled as follows:

$$C_{j,t+1} \sim NB\left(\lambda_{j,t}, \theta\right) \tag{1}$$

$$\log \lambda_{j,t} = \beta_0 + \beta_j \log\left(C_{j,t} + 1\right) + \log N_t \tag{2}$$

where $\beta_0$ is an intercept, seasonality is incorporated in the form of $\beta_j$ a multiplicative factor scaling transmission for biweek $t$ in biweek category $j \in \{1, \dots, 26\}$, $C_{j,t}$ is the number of probable dengue cases in biweek $t$, $N_t$ is the population size (specific to the year and location), and $\theta$ is the dispersion parameter. We did not fit a separate model for Vaupés, Colombia, since there are no probable dengue case counts reported for this department in 2007, 2008, 2009, 2011, and 2015. We also did not fit any regression models for the capital of Colombia, Bogotá, since there is limited corresponding dengue incidence and these cases are considered to originate from other departments.

**Fitting procedure and model performance.** We fit all time series models using the rstanarm R package[44] by implementing Bayesian MCMC methods. For each model, we sampled four chains with 10,000 iterations each (5,000 iterations

included as warmup) for subnational level models and 10,000 iterations each for national models. Convergence was evaluated by using the launch_shinystan function of the rstanarm R package[44]. We primarily assessed the convergence of our models using the Gelman-Rubin convergence statistic[45] and deemed convergence adequate when $\hat{R}$ is less than 1.1. We also checked to see whether there were any parameters with an effective sample size less than 10% of the total sample size or any parameters with a Monte Carlo standard error greater than 10% of the posterior standard deviation. We noted whether there were any divergent transitions after the warmup period. We evaluated model performance by calculating $R^2$ values for the focal year of predictions (out-of-sample values) and predictions of the data used to fit the models (in-sample values) (see Supplementary Fig. 3).

**Comparisons between predicted and observed dengue incidence.** Starting with the second biweek of data for a given location, we sampled 500 values from the posterior distribution for predicted incidence for the corresponding model (fit using data from all other years). We evaluated the median prediction for each biweek and visually compared this value to the number of observed dengue cases in that biweek (Supplementary Fig. 4). We then evaluated the quantile of the observed incidence in that biweek in the cumulative distribution of posterior predicted values. We consider the observed state- or department-level incidence in a particular biweek to be statistically atypical if it falls outside of the 90% prediction interval (PI), i.e., if the observed quantile is less than 0.05 or greater than 0.95 (see Fig. 2 and Supplementary Fig. 5). We repeated this analysis using a Bonferroni adjusted quantile (Supplementary Fig. 6).

Further, we implemented a permutation test to consider whether the number of atypically high or low observed incidence values in each year (separately for each country) was significant. For each location, we reassigned the years (sampling without replacement). Then for each year we counted the number of statistically high or low values of observed incidence. We repeated this procedure 10,000 times and then found the quantile of the observed numbers of atypically high or low biweeks within the cumulative distribution function generated from the permuted data (see Fig. 2a, c). This permutation procedure preserves temporal correlation within the years. We considered a second permutation test that preserved spatial correlation within each biweek. For this test, we reassigned biweek labels. For each biweek, we sampled without replacement from the years of available predictions for the corresponding biweek category (ranging from 1–26). We then reassigned the quantiles of that particular biweek to be the corresponding incidence from that biweek category in the resampled year. Again, we performed 10,000 permutations and found the quantile of the observed counts of statistically high or low biweeks in the corresponding cumulative distribution function. Results were similar between the two permutation tests. Between the two tests, quantile differences for each year were less than 5% (median value 0.012, $n = 58$) and significant results ($q > 95\%$) presented in Fig. 2 were maintained. Note that we did not include 1999 in the permutation test for Brazil since data is not available for the entire year in Acre (the first dengue case in the dataset is in August 1999).

**Hierarchical regression models.** Separately for Brazil and Colombia, we fit a set of spatial hierarchical models for dengue incidence (using state-level data for Brazil models and department-level data for Colombia models) with negative binomial errors. We considered models with either a log-additive effect for recent years (2015, 2016, 2017) or a multiplicative effect of either Zika or chikungunya case counts (previous biweek or total count recorded up to and including the previous biweek) on expected dengue cases. We focus on an absolute incidence version of the year effect model, where a dengue seasonality (biweek) indicator is used instead of the log-dengue case count predictor used in the other models. The subnational location-specific effects account for deviation from the country-level shared effects. During the fitting procedure, a variance term is also fit for the distribution of location-specific effects. These models take the following form:

$$C_{i,j,t+1} \sim NB\left(\lambda_{i,j,t}, \theta\right) \tag{3}$$

Additive model:

$$\log \lambda_{i,j,t} = \beta_0 + \beta_{i,j} + \alpha_{\text{year}(t)} + \alpha_{i,\text{year}(t)} + \log N_{i,t} \tag{4}$$

$$\alpha_{i,\text{year}} \sim N\left(0, \sigma_{\text{year}}\right) \tag{5}$$

Multiplicative model:

$$\log \lambda_{i,j,t} = \beta_0 + \beta_{i,j} \log\left(C_{i,j,t} + 1\right) + (\alpha + \alpha_i)\log(X_{i,t} + 1) + \log N_{i,t} \tag{6}$$

$$\alpha_i \sim N(0, \phi) \tag{7}$$

In the additive model, $\alpha_{\text{year}}$ represents the shared effect for 2015, 2016, 2017. $\alpha_{i,\text{year}}$ and $\alpha_i$ are location-specific (subnational) terms for the additive year effect and multiplicative arbovirus-related incidence effect respectively (with dispersion parameters $\alpha_{\text{year}}$ and $\phi$). There is one intercept coefficient ($\beta_0$) and one coefficient for each state biweek pair ($\beta_{i,j}$), where $i$ ranges over the subnational locations and $j$ ranges over the 26 biweeks. These coefficients allow for seasonal differences in transmission intensity at the subnational level. Additionally, there is an offset for year-specific subnational population size ($N_{i,t}$). $X_{i,t}$ represents either:

Zika or chikungunya incidence (either at biweek $t$ or the cumulative total number of cases reported up through biweek $t$ in location $i$). We consider year and arbovirus-related incidence coefficients to be significant when their 95% Bayesian credible interval (CrI) does not overlap zero.

**Stochastic compartmental model.** We developed a four serotype DENV model that also allowed for ZIKV infection in both mosquito and human populations. There are six mosquito compartments: susceptible, infected with one of the four dengue serotypes, and infected with ZIKV. The human compartments are based on both DENV and ZIKV status. With respect to DENV infection status, individuals in the human population are either susceptible, infectious, cross-protected, or recovered. We keep track of primary and secondary DENV infections and assume that after two heterotypic DENV infections, individuals become immune to further DENV infections. Homotypic reinfections are not allowed in the model. The human classes are further stratified depending on the individual's ZIKV status: either susceptible, infectious, cross-protected (from DENV), or recovered. There are 125 total compartments in the model.

We considered a suite of immune-mediated interaction scenarios, considering the possibility of enhancement or cross-protection. The enhancement scenarios we considered included: no enhancement, DENV enhances DENV, DENV enhances DENV and ZIKV, and enhancement in all directions. Enhancement was incorporated as a 1.6-fold increase in the force of infection from humans to mosquitoes. We incorporated a symmetric cross-protective effect, i.e. for a period of time, a dengue infection would reduce the hazard of a future Zika infection and a Zika infection would reduce the hazard of a future dengue infection. Cross-serotype dengue protection (80% hazard reduction to other DENV serotypes) was included in all scenarios. The average duration of cross-protection between DENV serotypes was one year ($\gamma = 1$)[46]. We considered the case when cross-protection between dengue and Zika also lasted one year on average. The scenarios without cross-protection did allow for a reduced hazard for approximately 4 days ($\gamma = 1/100$). We considered two levels of cross-protection between DENV and ZIKV. The high-level matched the level of cross-serotype dengue protection ($\rho = 0.2$)[46]. The low-level of cross-protection reduced the hazard for the other virus by only 20% ($\rho = 0.8$). We did not incorporate seasonality in either DENV or ZIKV transmission but expect simulations with seasonality to be qualitatively similar to those presented here.

We ran 100 simulations for each scenario using a tau-leap approximation of the Gillespie method in Fortran. ZIKV was introduced into the mosquito population after 100 years of DENV only simulations. Numerical solutions for the corresponding deterministic system (using randomized initial conditions) simulated for a period of 600 years (to approximate steady state) were used to determine the initial conditions for the stochastic simulations mentioned here. An additional 20 years of data was simulated after the introduction of ZIKV. We considered DENV and ZIKV reproduction numbers for all combinations of 2 and 4 (see Fig. 4 for the case when DENV and ZIKV reproduction numbers are 4 and 2 respectively), and for the case when both are set to 3 (see Supplementary Fig. 10). The human population size was 10 million and the mosquito population size was 20 million (ratio of total mosquito population to human population set to 2)[47]. The human death rate was 0.02 per year[48] and the mosquito death rate was 15 per year[49]. Simulation data analysis was performed in R.

We also considered the case when ZIKV was introduced after 40 years of DENV only simulations (see Supplementary Figs. 11 and 12). For these simulations, introduction of DENV1-4 infected mosquitoes are introduced according to Poisson processes with rate parameter such that on average three infected mosquitoes are introduced each year for each serotype. Sample simulations displaying DENV dynamics prior to the introduction of ZIKV under both stable and recently introduced DENV conditions are displayed in Supplementary Fig. 12.

To evaluate the effects of incorporating cross-protection or enhancement, we consider changes in trough duration and peak size. All scenarios that incorporated an immune-mediated interaction were evaluated against the baseline determined by the scenario with no enhancement and no cross-protection between DENV and ZIKV (see Fig. 4a). We define the trough duration to be the length of time that aggregated DENV prevalence is consecutively less than one half of the average incidence following the introduction of ZIKV, in the baseline scenario (approximately 16 individuals per 100,000 population). We set the trough duration to zero for all simulations that did not include a complete trough within the 20 years following the introduction of ZIKV. Peak size is defined to be the maximum DENV prevalence value, aggregated across all serotypes, in the 20 years following the introduction of ZIKV. Changes in peak size resulting from incorporating cross-protection between DENV and ZIKV were evaluated by dividing the average peak size for scenarios without enhancement by the average peak size in the baseline case (where the average is taken across the 100 simulations of each scenario). Peak size and trough duration averages and inter-quantile ranges across the 100 simulations of each scenario are displayed in Supplementary Table 1.

**Reporting summary.** Further information on research design is available in the Nature Research Reporting Summary linked to this article.

## Code availability

The code used to process the case count data and to produce our results is available in GitHub repository UF-IDD/dengue-Zika-chik_Americas [https://github.com/UF-IDD/dengue-Zika-chik_Americas] (ref. [25]).

## Data availability

The case count and population data gathered and analyzed in this study are available in GitHub repository UF-IDD/dengue-Zika-chik_Americas [https://github.com/UF-IDD/dengue-Zika-chik_Americas] (ref. [25]).

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

## Acknowledgements

We thank S. E. Bellan, M. A. Johansson, and members of the U.F. and J.H.U. Infectious Disease Dynamics groups for helpful discussions. This research was supported by the National Institute of Allergy and Infectious Diseases at the National Institutes of Health (grant R01AI102939).

## Author contributions

R.K.B., A.H., S.D.M., and G.D.K. synthesized the data; R.K.B., A.H., S.D.M., and S.C.C. analyzed the data; R.K.B. fit the statistical models with assistance from J.L. and D.A.T.C.; L.M.R. performed the simulations, L.M.R. and R.K.B. analyzed the simulation results; R.K.B., A.H., D.P.R., I.R.-B., L.M.R., J.L., and D.A.T.C. interpreted the results; R.K.B., A.H., and D.A.T.C. wrote the original draft; L.C.K., D.P.R, I.R.-B., L.M.R., J.L., and D.A.T.C. contributed written feedback. All authors wrote the manuscript.

## Competing interests

The authors declare no competing interests.
