## [Peer Review File · Nature Communications]

Reviewers' comments:

Reviewer #1 (Remarks to the Author):

Borchering and colleagues investigate an interesting, and potentially important, aspect of dengue epidemiology post the Zika epidemic – is Zika somehow the cause of the lower than normal dengue cases? This topic of potential protection or enhancement during subsequent infections of these viruses has been recently debated in the literature, and more epidemiological and modeling studies like this are certainly needed to help us understand the underlying interactions. The data and methods are appropriately used, and the results and discussions are clearly presented.

Importantly, because the results are mixed do not fully answer the question of “what happened to dengue?” (likely because of the data), the author's did a nice job of not over interpreting to fit any a priori hypotheses (if they had any) and outline how future studies can build upon this. Moreover, their results suggest that dengue incidence will not stay low, but rather some regions may be susceptible to large dengue epidemics in the near future (which, based on some surveillance data, may be correct). I do not have any major reservations that would prevent this work from being published, only minor suggestions to help improve the manuscript.

Minor suggestions:

1) Title: I personally prefer major conclusions as titles, not an overview of what was done. As this is a preference, I would not normally bring this up, but it may be something that the authors should consider here. Because of the debate in the literature between Zika-dengue enhancement/protection, some may see the title and cherry pick a finding out of the abstract to fit their a priori hypotheses for either protection or enhancement. If the title explicitly stated the uncertainty of prior Zika outbreaks on the local declines in dengue cases (we'll leave it up to the authors to word this), it may help to ensure that their work gets cited correctly.

2) Introduction (line 41): The authors state that dengue and Zika “viruses may interact competitively or synergistically through the human immune responses” and later describe these in some detail. While I agree that neutralizing (or not-quite-neutralizing) antibodies are likely the key factor here, they are not the only possible way for competition or synergism. If subsequent infections occur during the acute phase, the interferon or other innate immune defenses may also play a prominent role. In fact, many co-infections were reported during the Zika epidemic. In addition, competition may also occur in mosquitoes (though based on the low infection rates, not as likely). And finally, a third hypothesis was not discussed, that there is neither protection or enhancement (i.e. neutral). We recently published a perspective on this subject that may help the authors briefly describe other ways that these viruses may interact – which may be epidemiological important

(<https://journals.plos.org/plosbiology/article?id=10.1371%2Fjournal.pbio.3000130>). [Please note that I am not “fishing” for a citation, I'm only trying to provide a reference for additional points to think about. The primary literature would be more important to cite anyways.]

3) Data and code availability: It would be very helpful if the curated raw data and the codes (R and Matlab) used in this manuscript could be shared on public repositories for others to follow up on this work.

I no longer participate in blind review – Nathan Grubaugh (with the help from a trainee).

Reviewer #2 (Remarks to the Author):

This paper addresses an important and timely question, the interaction between two arboviruses of global concern, with data on two large countries of South America, Brazil and Colombia. Approaches include statistical models (time series and hierarchical regression models) and

process-based dynamical models. The main (and only consistent) result shows anomalous low incidence of dengue in 2017 at the national level for both countries following the emergence of Zika. Results from data at higher spatial resolutions were more mixed, with inconsistent directions of the effects in time and space. Although I appreciate the effort and value of analyzing extensive surveillance data, I have a number of major concerns on the logic and methods. Reporting error may make it difficult to reach clearer conclusions but it is difficult to tell whether the mixed conclusions have more to do with limitations of the analyses themselves.

Major comments:

- 1) The emergence of the Zika virus happened in Brazil following the sequential invasion of the four serotypes of dengue. This context for the effect of Zika on dengue does not seem to be addressed in the formulation of the analyses. For example, why do the simulations of the process-based models consider the long-term dynamics after removing transients?. This is an unrealistic assumption for the timing of Zika's emergence. The seasons of emergence and re-emergence of dengue should matter also for the statistical models. In Brazil, the four serotypes emerged sequentially since 1986, with large seasonal outbreaks that lasted for two to three years depending on location, with other intermittent re-emergent outbreaks in between low seasons.
- 2) Predictions of the statistical models were used to identify departures from these predictions in observations that were then interpreted as atypical. This approach requires a demonstration that a given model predicts and fits the data well. Otherwise, deviations from predictions could also mean that the models do not capture the dynamics of dengue sufficiently well.
- 3) The model structure of the statistical models, especially the time series ones, needs better explanation. Why was a first-order autoregressive model chosen? Why not a seasonal autoregressive model? Why does it make sense to look at atypical low or high incidence at the level of bi-weeks? Wouldn't one expect the effect of Zika on dengue to operate over the season in a systematic way and not to flip directions at that temporal scale?
- 4) The description of the dynamical models is limited. There is no description of how the models were parameterized.
- 5) It is not clear that the conclusions reached with the dynamical models do require the models. The general findings from simulations seem to be what one would expect. High dengue years should occur after some number of low seasons from the observed population dynamics of dengue, and from its SIR-type dynamics, in this part of the world.

Reviewers' comments:

Reviewer #1 (Remarks to the Author):

Borchering and colleagues investigate an interesting, and potentially important, aspect of dengue epidemiology post the Zika epidemic – is Zika somehow the cause of the lower than normal dengue cases? This topic of potential protection or enhancement during subsequent infections of these viruses has been recently debated in the literature, and more epidemiological and modeling studies like this are certainly needed to help us understand the underlying interactions. The data and methods are appropriately used, and the results and discussions are clearly presented. Importantly, because the results are mixed do not fully answer the question of “what happened to dengue?” (likely because of the data), the author’s did a nice job of not over interpreting to fit any a priori hypotheses (if they had any) and outline how future studies can build upon this. Moreover, their results suggest that dengue incidence will not stay low, but rather some regions may be susceptible to large dengue epidemics in the near future (which, based on some surveillance data, may be correct). I do not have any major reservations that would prevent this work from being published, only minor suggestions to help improve the manuscript.

Minor suggestions:

1) Title: I personally prefer major conclusions as titles, not an overview of what was done. As this is a preference, I would not normally bring this up, but it may be something that the authors should consider here. Because of the debate in the literature

between Zika-dengue enhancement/protection, some may see the title and cherry pick a finding out of the abstract to fit their a priori hypotheses for either protection or enhancement. If the title explicitly stated the uncertainty of prior Zika outbreaks on the local declines in dengue cases (we'll leave it up to the authors to word this), it may help to ensure that their work gets cited correctly.

We appreciate the suggestion, but struggled to find a title that conveyed the mix of results that we describe and so think the present title is consistent with a manuscript that presents multiple contrasting findings rather than a single conclusion.

2) Introduction (line 41): The authors state that dengue and Zika “viruses may interact competitively or synergistically through the human immune responses” and later describe these in some detail. While I agree that neutralizing (or not-quite-neutralizing) antibodies are likely the key factor here, they are not the only possible way for competition or synergism. If subsequent infections occur during the acute phase, the interferon or other innate immune defenses may also play a prominent role. In fact, many co-infections were reported during the Zika epidemic. In addition, competition may also occur in mosquitoes (though based on the low infection rates, not as likely). And finally, a third hypothesis was not discussed, that there is neither protection or enhancement (i.e. neutral). We recently published a perspective on this subject that may help the authors briefly describe other ways that these viruses may interact – which may be epidemiological important (<https://journals.plos.org/plosbiology/article?id=10.1371%2Fjournal.pbio.3000130>). [Please note that I am not “fishing” for a citation, I’m only trying to provide a reference for additional points to think about. The primary literature would be more important to cite anyways.]

We agree that we have not expressed all the ways that these viruses could interact. We have expanded our introduction (p2) to state:

“Evidence suggests these viruses may interact competitively or synergistically through human immune responses: via antibodies in the case of non-overlapping infections^{4–10} or innate defenses during co-infections [Badolato-Corrêa J et al. 2018; Vogels et al. 2019]. Concurrent infections in the vector could also potentially alter viral fitness though the low prevalence of infection in mosquitos at any time and thus the low rate at which concurrent infections occur is likely to minimize the impact of this interaction [Chaves et al. 2018; Ruckert et al. 2017; Vogels et al. 2019]. It is also possible that these viruses may also have no biological interaction whatsoever.”

3) Data and code availability: It would be very helpful if the curated raw data and the codes (R and Matlab) used in this manuscript could be shared on public repositories for others to follow up on this work.

We have posted the curated raw data and our code in a Github repository (Github: UF-IDD/dengue-Zika-chik_Americas) (DOI 10.5281/zenodo.2566509).

I no longer participate in blind review – Nathan Grubaugh (with the help from a trainee).

We are very appreciative of the review.

Reviewer #2 (Remarks to the Author):

This paper addresses an important and timely question, the interaction between two arboviruses of global concern, with data on two large countries of South America, Brazil and Colombia. Approaches include statistical models (time series and hierarchical regression models) and process-based dynamical models. The main (and only consistent) result shows anomalous low incidence of dengue in 2017 at the national level for both countries following the emergence of Zika. Results from data at higher spatial resolutions were more mixed, with inconsistent directions of the effects in time and space. Although I appreciate the effort and value of analyzing extensive surveillance data, I have a number of major concerns on the logic and methods. Reporting error may make it difficult to reach clearer conclusions but it is difficult to tell whether the mixed conclusions have more to do with limitations of the analyses themselves.

Major comments:

1) The emergence of the Zika virus happened in Brazil following the sequential invasion of the four serotypes of dengue. This context for the effect of Zika on dengue does not seem to be addressed in the formulation of the analyses. For example, why do the simulations of the process-based models consider the long-term dynamics after removing transients?. This is an unrealistic assumption for the timing of Zika's emergence. The seasons of emergence and re-emergence of dengue should matter also for the statistical models. In Brazil, the four serotypes emerged sequentially since 1986, with large seasonal outbreaks that lasted for two to three years depending on location, with other intermittent re-emergent outbreaks in between low seasons.

We agree that the simulations with sequential introduction of dengue followed by a Zika outbreak 40 years after the re-introduction of dengue in the simulated population is an interesting set of simulations to consider given the history of dengue re-introduction in Brazil and Colombia and have created another set of simulations to explore this.

While we feel that a detailed reconstruction of these histories is beyond the scope of this manuscript, we have instead performed two sets of simulations to represent a spectrum of states of the dengue system when Zika was introduced. We present our original simulations that assume that dengue transmission was stable when Zika virus

was introduced as one of the two. We feel that this is justified due to the fact that dengue had circulated in the Americas up until a two to three decade absence in the middle of the 20th century. Though dengue was re-introduced in the 1980's, significant immunity existed in the population among those that were alive in the 1960's when dengue was present. We think that this level of immunity could contribute to the system being in a stable state.

In a new set of simulations included in the manuscript, we introduce Zika 40 years after dengue was introduced and explore a stochastic set of realizations of the possibly sequential introduction of dengue serotypes. These simulations do not assume that dengue was in a stable state when Zika was introduced.

In this set of simulations, we repeat all model formulations (regarding the inclusion of enhancing and cross-protective interactions between dengue and Zika) using simulations in which dengue viruses are introduced to the population over 40 years stochastically in a way that is consistent with the observed timing of detection in Brazil over the last 40 years, and then introducing Zika [Messina et al. 2014]. Our simulations introduce all four serotypes of dengue at year 0, then have a small risk of introduction of each serotype at each time point since then. These simulations create patterns of introduction consistent with historic patterns of introduction to Brazil, but model a broader pattern of possible introductions and thus do not rely on poorly observed information on when serotypes appeared in Brazil. We modeled 100 different introduction scenarios for each parameter combination considered and compared these results by multiple metrics to those in the original submission.

Supplementary Fig. 11 presents simulated time series from models where Zika is introduced after a sequential introduction of dengue. Supplementary Table 1 presents a comparison of the results for these simulations and the original simulations which introduced Zika to a population in which all four dengue serotypes had circulated for over 100 years. The results are qualitatively similar, though there are some quantitative differences. In general, the inclusion of the sequential introduction of dengue serotypes into our simulations leads to dengue being in more of a transient state when Zika is introduced, leading to greater variability in dengue dynamics after Zika is introduced. Lengths of times for which dengue is below a threshold were in general longer and peaks in dengue subsequent to Zika outbreaks higher in simulations with sequential historic introductions of dengue than simulations where dengue was in a stable state, though there was high variance in these findings.

We include results in the supplement (Supplementary Fig. 11, Table 1) and text in the revised manuscript (see below) describing these simulations.

Based on these updates, we made a few minor changes to the reported values for peak increases in the main text. We identified and corrected a few typos (highlighted in the

revised text). We also added the following text at the end of our Conclusion section (p8) to reflect recent observations of high dengue case counts in Brazil and Colombia:

“As of August, 2019, both Brazil and Colombia have reported more cases at this point in the year than in all of 2017 and 2018, though their dengue seasons are not complete. These indications are consistent with our expectations of the impact of higher levels of susceptibility due to lower incidence in recent years.”

2) Predictions of the statistical models were used to identify departures from these predictions in observations that were then interpreted as atypical. This approach requires a demonstration that a given model predicts and fits the data well. Otherwise, deviations from predictions could also mean that the models do not capture the dynamics of dengue sufficiently well.

We have added Supplementary Fig. 4 to visualize the performance of our statistical model in predicting and fitting the observed data. In this figure, we plot predicted vs. observed subnational incident cases per biweek for both Brazil and Colombia. We present R^2 values for the subnational models and have added the following statement to the main text (p4):

“...(see Methods and Supplementary Fig. 2-4 for further details on model implementation and predictive ability).”

In the Methods (p17), we have added:

“We evaluated the median prediction for each biweek and visually compared this value to the number of observed dengue cases in that biweek (Supplementary Fig. 4).”

3) The model structure of the statistical models, especially the time series ones, needs better explanation. Why was a first-order autoregressive model chosen? Why not a seasonal autoregressive model? Why does it make sense to look at atypical low or high incidence at the level of bi-weeks? Wouldn't one expect the effect of Zika on dengue to operate over the season in a systematic way and not to flip directions at that temporal scale?

We incorporated a reference to Finkenstadt and Grenfell 2000 to support our choice of a first-order autoregressive model (p16):

“Time series models. For each year of available data, we fit a *seasonal* one-step autoregressive model⁴³ with negative binomial errors for each state in Brazil and each department in Colombia using incidence data from that location in all other years.”

We thank the reviewer for pointing out that the description of our time series models needs clarification. We incorporated seasonality in our autoregressive model by estimating a separate transmission intensity coefficient for each biweek of the year for each subnational location. These biweekly coefficients repeat each year thus presenting

effectively a seasonal autoregressive model. There are certainly other ways to implement seasonality in autoregressive models, but we felt that this method was appropriate given the heterogeneous seasonality (particularly across Brazil) and linked cases in the last biweek to the next in a seasonally varying way. To highlight the connection between seasonality and our time series models we have added the following text (p16): “...seasonality is incorporated in the form of β_j ...”. Additionally, in the hierarchical model section we have added (p19): “These coefficients allow for seasonal differences in transmission intensity at the subnational level.”

We chose to consider models of biweekly incidence in order to make the most of the temporal scale of data available. We agree that one wouldn't expect the effect of Zika on dengue to fluctuate from biweek to biweek, and it was not our intention to try to detect such changes. We also feel that detecting changes to biweekly autoregressive terms provides a more stringent test to detect differences in dengue incidence. Effectively, by looking at biweek to biweek case numbers, we are adjusting for the level of dengue in each biweek and thus not simply detecting low dengue case numbers because previous weeks were lower. Our question is, given a certain number of cases at a particular time of year, do we see an expected number of cases in the next biweek (as would arise from transmission from those cases) or do we see a lower or higher amount.

The reviewer makes a valuable point that it is important to consider the time scale in which we would expect potential effects of ZIKV introduction to operate. Immune-mediated interactions in particular likely would take longer than a biweek to become established. This was one reason that we chose to also consider the effect of cumulative Zika incidence rather than biweekly incidence alone.

4) The description of the dynamical models is limited. There is no description of how the models were parameterized.

We have included a more detailed description of our dynamical models in the main text, Methods, and Supplementary Figures.

In the Stochastic simulations incorporating immune-mediated interactions section of the main text (p6-7) we now direct the reader to see the Methods section for further details and have added some details on the immune-mediated scenarios under consideration:

“We used a stochastic compartmental model that incorporated combinations of cross-protection or enhancement between the two viruses (*see Methods for further details*). We performed simulations in which ZIKV was introduced to a population in which DENV was in a stable state as well as simulations that incorporated the sequential introduction of dengue over the decades preceding the ZIKV introduction reflecting the observed detection of DENV serotypes³⁸. In simulations where ZIKV infection temporarily reduces an individual's risk of DENV infection by 80%, Zika epidemics are followed by a trough

in dengue incidence ranging from 2.2 years to 3.5 years depending on the *enhancement* scenario (Fig. 4i-l and Supplementary Table 1). *Even in the absence of enhancement between DENV and ZIKV, multiple simulations showed increases in dengue after troughs ranging from a 1.3-fold increase to a 2.7-fold increase (Fig. 4).*”

In the Stochastic compartmental model section of the Methods, we have added the following text describing the parameterization of the dynamical models (p20):

“Enhancement was incorporated as a 1.6-fold increase in the force of infection from humans to mosquitos.”

References for parameter values were added for:

- Average duration of cross-protection between DENV serotypes [Reich et al. 2013] (p20)
- Level of cross-serotype dengue protection [Reich et al. 2013] (p20)
- Ratio of total mosquito population to human population [Focks et al. 2000] (p21)
- Human death rate [Ferguson et al. 2009] (p21)
- Mosquito death rate [Trpis et al. 1995] (p21)

We have also provided further description of the initial conditions for the original simulations (p20-21):

“Numerical solutions for the corresponding deterministic system (using randomized initial conditions) simulated for a period of 600 years (to approximate steady state) were used to determine the initial conditions for the stochastic simulations mentioned here.”

In the Fig. 4 legend (p12), we now direct readers to the Methods for further details and have added the following text:

“DENV and ZIKV reproduction numbers are assumed to be 4 and 2, respectively. Other reproduction number combinations are considered in Supplementary Fig. 10). The dashed line indicates one-half of the average incidence in panel a which we use to define the start and end of DENV prevalence troughs (see Methods and Supplementary Table 1).”

For the new simulations (discussed above in Major Comment 1), we have added (p21):

“We also considered the case when ZIKV was introduced after 40 years of DENV only simulations (see Supplementary Fig. 11). For these simulations, introduction of DENV1-4 infected mosquitoes are introduced according to Poisson processes with rate parameter such that on average three infected mosquitoes are introduced each year for each serotype.”

We have also added detail in the legend for Supplementary Fig. 11:

“Simulation results incorporating immune-mediated interactions between DENV and ZIKV. Mean and 95% inter-quantile range from stochastic simulations spanning 10 years post ZIKV-introduction. 100 simulations per scenario (see Methods for further details). ZIKV introduced 40 years after DENV is introduced in a susceptible population. Mosquitos infected with each of the four DENV serotypes are introduced as a Poisson process with an average of three per year. DENV and ZIKV reproduction numbers are 4 and 2 respectively. Other reproduction number combinations are considered in Supplementary Fig. 10). The dashed line indicates one-half of the average incidence in panel a which we use to define the start and end of DENV prevalence troughs. a-d Individuals with previous dengue exposure experience 20% of the DENV force of infection (FOI) that a fully susceptible person would. e-h Individuals with previous ZIKV exposure experience 80% of the FOI that a fully susceptible person would. i-l Individuals with ZIKV exposure experience 20% of the DENV FOI (same amount of cross-protection between dengue and Zika than between dengue serotypes).”

5) It is not clear that the conclusions reached with the dynamical models do require the models. The general findings from simulations seem to be what one would expect. High dengue years should occur after some number of low seasons from the observed population dynamics of dengue, and from its SIR-type dynamics, in this part of the world.

We agree with the reviewer’s observation that high dengue years would naturally follow periods of low dengue incidence in this system. However, we would like to highlight that our models explore more fully the expected behavior under a range of scenarios of possible interaction between dengue and Zika. We also found a counter-intuitive result. If dengue and Zika reproduction numbers are correlated spatially, as we expect them to be, then we would not expect to find an association between the amount of Zika a location has had and the reduction in dengue incidence the location experiences even when we assume Zika provides a period of cross-protection. We hypothesized that places with more Zika would have greater reductions in dengue. We did not observe this association. The dynamical models give us intuition for why this is the case (see Supplementary Fig. 10). Areas with higher amounts of Zika would also experience higher transmission intensity of dengue. Simulations with higher reproduction numbers for dengue have greater robustness in their dengue incidence to cross-protective effects of Zika, with dengue incidence remaining more consistent temporally before and after Zika outbreaks.

New References:

Badolato-Corrêa, J. *et al.* Human T cell responses to Dengue and Zika virus infection compared to Dengue/Zika coinfection. *Immunity, inflammation and disease* **6**, 194–206 (2018).

Chaves, B. A. *et al.* Coinfection with Zika Virus (ZIKV) and Dengue Virus Results in Preferential ZIKV Transmission by Vector Bite to Vertebrate Host. *J. Infect. Dis.* **218**, 563–571 (2018).

Ferguson, N., Anderson, R. & Gupta, S. The effect of antibody-dependent enhancement on the transmission dynamics and persistence of multiple-strain pathogens. *Proc. Natl. Acad. Sci. U. S. A.* **96**, 790–794 (1999).

Finkenstädt, B. F. & Grenfell, B. T. Time series modelling of childhood diseases: a dynamical systems approach. *J. R. Stat. Soc. Ser. C Appl. Stat.* **49**, 187–205 (2000).

Focks, D. A., Brenner, R. J., Hayes, J. & Daniels, E. Transmission thresholds for dengue in terms of *Aedes aegypti* pupae per person with discussion of their utility in source reduction efforts. *Am. J. Trop. Med. Hyg.* **62**, 11–18 (2000).

Messina, J. P. *et al.* Global spread of dengue virus types: mapping the 70 year history. *Trends Microbiol.* **22**, 138–146 (2014).

Reich, N. G. *et al.* Interactions between serotypes of dengue highlight epidemiological impact of cross-immunity. *J. R. Soc. Interface* **10**, 20130414 (2013).

Rückert, C. *et al.* Impact of simultaneous exposure to arboviruses on infection and transmission by *Aedes aegypti* mosquitoes. *Nat. Commun.* **8**, 15412 (2017).

Trpis, M., Häusermann, W. & Craig, G. B., Jr. Estimates of population size, dispersal, and longevity of domestic *Aedes aegypti aegypti* (Diptera: Culicidae) by mark--release--recapture in the village of Shauri Moyo in eastern Kenya. *J. Med. Entomol.* **32**, 27–33 (1995).

Vega-Rúa, A., Zouache, K., Girod, R., Failloux, A.-B. & Lourenço-de-Oliveira, R. High level of vector competence of *Aedes aegypti* and *Aedes albopictus* from ten American countries as a crucial factor in the spread of Chikungunya virus. *J. Virol.* **88**, 6294–6306 (2014).

Vogels, C. B. F. *et al.* Arbovirus coinfection and co-transmission: A neglected public health concern? *PLoS Biol.* **17**, e3000130 (2019).

REVIEWERS' COMMENTS:

Reviewer #2 (Remarks to the Author):

The authors have addressed my comments and now the paper is clear on the methods, results and interpretations. The context for the predicted increase in dengue is also clearer, which now makes this conclusion interesting.

The addition of the new simulations with dengue's introductions (vs. the previous case of dengue at steady state) makes the result of an enhanced risk of a dengue epidemic more credible. I have one comment on this addition: it would be informative to include examples of the resulting dynamics of dengue incidence (including by serotype) before the Zika introduction in the Supplement. This would provide a better sense for the population dynamics of the disease pre-Zika arrival in the model.

It is also worth clarifying whether seasonality in the transmission rate was incorporated in the simulations. I may have mis-interpreted or miss this information in the model description. I am mistaken in thinking there was a constant R_0 and no seasonality? If so, it would be useful for the authors to clarify this.

Response to Referees:

Reviewer #2 (Remarks to the Author):

The authors have addressed my comments and now the paper is clear on the methods, results and interpretations. The context for the predicted increase in dengue is also clearer, which now makes this conclusion interesting.

The addition of the new simulations with dengue's introductions (vs. the previous case of dengue at steady state) makes the result of an enhanced risk of a dengue epidemic more credible. I have one comment on this addition: it would be informative to include examples of the resulting dynamics of dengue incidence (including by serotype) before the Zika introduction in the Supplement. This would provide a better sense for the population dynamics of the disease pre-Zika arrival in the model.

We agree that displaying dengue dynamics prior to ZIKV introduction is informative and thank the reviewer for this suggestion. Sample simulations are now displayed for DENV incidence 20 years prior to ZIKV introduction in Supplementary Fig. 12. The following legend accompanies this figure:

“SI Figure 12: Serotype-specific DENV incidence dynamics 20 years prior to the introduction of ZIKV. (a) ZIKV is introduced when DENV is in a stable state (time = 100). (b) ZIKV introduced 40 years after DENV is introduced in a susceptible population. Mosquitos infected with each of the four DENV serotypes are introduced as a Poisson process with an average of three per year. DENV and ZIKV reproduction numbers are 4 and 2 respectively.”

We have also incorporated a reference to this figure in the main text:

“We performed simulations in which ZIKV was introduced to a population in which DENV was in a stable state as well as simulations that incorporated the sequential introduction of dengue over the decades preceding the ZIKV introduction reflecting the observed detection of DENV serotypes³⁸ (see Supplementary Fig. 12 for sample simulations from both settings).”

It is also worth clarifying whether seasonality in the transmission rate was incorporated in the simulations. I may have mis-interpreted or miss this information in the model description. I am mistaken in thinking there was a constant R_0 and no seasonality? If so, it would be useful for the authors to clarify this.

We thank the reviewer for pointing out this ambiguity and have added the following statement to the Methods:

“We did not incorporate seasonality in either DENV or ZIKV transmission but expect simulations with seasonality to be qualitatively similar to those presented here.”